# Cell Mechanisms of Post-Mortem Excitability of Skeletal Muscle

**DOI:** 10.3390/biomedicines13010221

**Published:** 2025-01-17

**Authors:** H. Stigter, T. Krap, W. L. J. M. Duijst

**Affiliations:** Faculty of Law and Criminology, Maastricht University, Minderbroedersberg 4–6, 6211 LK Maastricht, The Netherlands; t.krap@maastrichtuniversity.nl (T.K.); wilma.duijst@maastrichtuniversity.nl (W.L.J.M.D.)

**Keywords:** cell mechanisms, post-mortem, excitability, skeletal, muscle

## Abstract

**Background/Objectives:** The excitability of skeletal muscle is a less-known post-mortem supravital phenomenon in human bodies, and it can be used to estimate the post-mortem interval. We conducted a field study in the Netherlands to investigate the applicability of muscle excitability (SMR) by mechanical stimulation for estimating the post-mortem interval in daily forensic practice. Knowledge concerning the post-mortem cell mechanisms accounting for the post-mortem excitability of skeletal muscle is lacking. Cell mechanisms are the specific intracellular and biochemical processes responsible for post-mortem muscle excitability. **Methods:** We have studied the theoretical backgrounds of the cell mechanisms that might be responsible for post-mortem muscle excitability, by performing literature research via the databank PubMed. **Results:** Based on the current available literature, in our opinion the intracellular changes in muscle cells that are responsible for SMR resemble the intracellular processes responsible for muscle fatigue due to energy exhaustion in the living. **Conclusions:** We hypothesize two pathways, depending on the level of energy in the muscle cell, that could be responsible for post-mortem muscle excitability by mechanical stimulation.

## 1. Introduction

Since time immemorial, it has been known that post-mortem changes take place after death and are influenced by body and environmental characteristics. Scientific investigation of these changes began in the eighteenth century and has been studied ever since [1]. These studies focussed primarily on the classical phenomena of body cooling (algor mortis), post-mortem lividity (livor mortis), and post-mortem muscle stiffness (rigor mortis). The essential goal of these studies was and has been up to now, to have a better understanding of the relation between the development of these post-mortem changes and the post-mortem interval (PMI) and, subsequently, to strive for a more accurate estimation of the time since death for crime scene investigation.

A less well-known post-mortem phenomenon is skeletal muscle excitability. Studies on the excitability of skeletal muscle were mainly conducted in Germany and published in German during the nineteenth century and the first half of the twentieth century [2,3,4,5,6]. A more recent study was conducted in 2012 by Warther et al. [7].

We conducted a field study in the Netherlands during a 7-year period (2017–2023) to investigate the applicability of muscle excitability by mechanical stimulation to estimate the PMI in daily forensic practice [8,9]. Recently, we have demonstrated that mechanical stimulation of the skeletal muscle of the arm is an easy-to-learn method for forensic physicians [8]. The outcome of mechanical stimulation of skeletal muscle, a so-called supravital muscle reaction (SMR) based on an idiomuscular contraction, can be of added value in daily forensic practice for estimating PMI, which was shown in recent case reports [9].

Though the post-mortem excitability of skeletal muscle is mentioned in textbooks [1] and used in daily forensic practice in the Netherlands, knowledge concerning the cell mechanisms of SMR is lacking. Cell mechanisms are the specific intracellular and biochemical processes responsible for post-mortem muscle excitability. The cell mechanisms in muscle cells responsible for muscle contraction in the living is well known and have been the subject of extensive in vitro animal studies [10,11,12,13,14,15,16,17,18,19,20,21,22,23,24]. The cell mechanisms in muscle cells responsible for post-mortem muscle contraction are not described in the literature. This gap in the current knowledge makes research and a theoretical reflexion on SMR necessary.

In the absence of physiological nerve stimulation as a trigger for muscle contraction post-mortem, it is also important to know which intracellular processes are responsible for causing skeletal muscles to contract and relax post-mortem after mechanical stimulation. After death, the metabolic capacity of cells will drop, and, accordingly, the level of glycogen and ATP in cells. SMR is dependent on the post-mortem storage of glycogen and adenosine triphosphate (ATP). Due to ATP deficiency and the influence of irreversible changes in the intracellular environment based on anaerobic glycolysis, which leads to a low intracellular pH, skeletal muscle cells lose their ability to contract, resulting in the loss of SMR. In this paper, we pose a hypothesis of the biomechanical processes responsible for SMR based on the contractile process in the case of living human beings.

## 2. Ante-Mortem Muscle Activity

Contraction of muscles in humans is a process that depends on calcium ions. Calcium is stored in the sarcoplasmic reticulum and is released under the influence of nerve stimulation. Muscle cells are mainly composed of actin and myosin filaments. Actin filaments consist of the protein actine, surrounded by tropomyosin and troponine in a helix-like structure. In the case of rest, i.e., no contraction of muscle, the tropomyosin component covers the active sites on the actin-binding parts. Troponin is a complex structure consisting of troponin-T that can bind to tropomyosin and troponin C, which serves as the binding place for calcium ions. Calcium ions, which are released into the sarcoplasm, bind to troponin C on the actin filaments. This leads to the displacement of tropomyosin covering the active sites on the actin filaments, exposing the binding sites and allowing the formation of crossbridges with myosin filaments via myosin heads, which is necessary for the process of muscle contraction. Crossbridges are formed based on the hydrolysis of ATP attached to the myosin head into adenosine diphosphate (ADP) and inorganic phosphate (Pi), producing energy necessary for crosslinking of the myosin head with an active part on the actin filament. Subsequently, ADP and Pi are separated from the myosin head, causing the myosin head to pivot and bend. This leads to the sliding of actine filaments along myosin filaments. Then, new ATP binds to the myosin head, which causes disconnection of the crossbridge, i.e., the myosin head separates from the actin filament. This is immediately followed by the conversion of ATP into ADP and Pi, leading to a new crosslink between the myosin head and another part of the actine component in the presence of calcium. The contractile cycle exists in different states, which means that it is an asynchronous process. Some of the crossbridges are active, and some are inactive or in another state of binding or unlocking. ATP-bounded myosin heads prevent the myosin head from binding with actin filaments. On other myosin heads, ATP is hydrolyzed in ADP and Pi, causing a weak binding between the myosin head and the actin filament, while at other places, there is a temporary state of strong binding between the myosin head and the actin filament where ADP and Pi are released from the myosin head [10,11,12,13,14,15,16,17,18,19,20,21,22,23,24].

The energy needed for this process of muscle contraction is provided by the breakdown of ATP and phosphocreatine. ATP is reproduced by anaerobic glycolysis (conversion of glucose into lactate) and oxidative phosphorylation. ATP is mainly consumed during repetitive attachment and detachment of myosin crossbridges at binding sites on adjacent actin filaments. ATP is also used for actively pumping back calcium into the sarcoplasmic reticulum, and the active pumping of ions sodium and potassium ions across the cell membrane, which are fundamental for the generation of action potentials. The fuel needed to produce ATP originates from carbohydrates, fatty acids, and triglycerides present in muscles. When nerve stimulation fades, the transport of calcium into the sarcoplasm stops while the active pumping back of calcium into the sarcoplasmic reticulum continues. This leads to the deactivation of the active sites on the actine filaments, and the muscle returns to its relaxed ground state [10,11,12,13,14].

## 3. Post-Mortem Muscle Excitability

When brain activity shuts down (clinical death), nerve stimulation of skeletal muscle stops. However, there is still energy available in muscle cells in terms of glycogen and ATP, while calcium ions are also still present in the muscle cells. Intracellular processes in muscle cells, such as the pumping of ions across membranes through ion channels, do not stop when the brain of the person shuts down at the moment of death. Thus, muscle cells, like other cells in the body, remain vital until exhaustion of energy and post-mortem decrease of the intracellular pH due to anaerobic glycolysis and build-up of carbon dioxide (right shift in the acidic base equilibrium) leads to death of the muscle cell. This period of temporary survival of muscle cells after brain death is fundamental for the post-mortem excitability of skeletal muscle.

After clinical death, as long as ATP is available, calcium is still actively pumped back into the sarcoplasmic reticulum. The intracellular calcium concentration decreases, which leads to the deactivation of active sites on the actine filaments. Myosin heads detach from the actin filament, while other myosin heads cannot bind to actin because of the presence of ATP on the myosin head. Other crossbridge formations are weakly bounded crossbridges (ATP hydrolyzed in ADP and Pi) that detach because of a shortage of calcium ions, whereas there are also strongly bounded crossbridges, remaining in a transient rigor state (myosin heads without ADP and Pi) despite the presence of a lower concentration of calcium ions.

Post-mortem mechanical stimulation of skeletal muscle can result in a visible and manually detectable contraction and relaxation. As described by Dotzauer et al. and observed in daily practice, dependent on the PMI, the strength, magnitude, and velocity of post-mortem contraction and relaxation of skeletal muscle observed after mechanical stimulation can be divided into three levels of degree. In the first 1.5 to 2.5 h after death, a contraction of the whole muscle can be observed, which corresponds with the muscle contraction observed by Zsako in the 1910s [3]. This is followed by a strong and reversible idiomuscular pad (4–5 h post-mortem; Figure 1). At the third level, only a weak idiomuscular pad can be observed, and it may persist for a long period [5,25,26]. Furthermore, longer PMI autolysis based on tissue decay by enzymatic degradation processes undermines the ability of muscle to contract due to loss of functional tissue [27,28]. Finally, muscle contraction can no longer be observed after mechanical stimulation.

An interesting and promising field of scientific development is the area of forensic proteomics, in which proteomic analysis of post-mortem skeletal muscle can provide insights into the biochemical changes that influence muscle excitability after death. Marrone et al. described nine proteins showing a decreasing or increasing trend of concentration in the muscle cell with increasing PMI. Among the proteins decreasing with longer PMI is the so-called Muscle-restricted coiled-coil protein (MURC), which is present in the sarcomere of skeletal muscle. MURC is involved in the myogenesis of skeletal muscle cells [29].

## 4. Hypothesis

When skeletal muscle is mechanically stimulated during the early hours post-mortem, calcium ions at the location where the muscle was hit flow into the muscle cell. This influx of calcium ions can be indirect because of local depolarisation of the cell membrane, comparable with the situation in the living, or direct due to rupture of the membrane of the sarcoplasmic reticulum. Calcium ions trigger the active binding sites of actin. The parts of the sarcomere with myosin heads containing ADP and Pi bind immediately with actin. The same accounts for ATP-bound myosin heads that are immediately hydrolyzed in ADP and Pi and additionally bind to actin. The parts that were in a transient rigor state detach from the actin because ATP is actively bound to the myosin head, causing relaxation of that part of the sarcomere. So, we believe that the different states of the contraction cycle characteristic for the situation in the living also account for the situation during the early hours post-mortem.

According to Cannon and Rosenblueth, a denervated structure becomes hypersensitive to neurotransmitters like acetylcholine (Ach) due to a decrease in the activity of the enzyme cholinesterase [30,31]. In the 1920s, Frank showed an increase in muscle excitability by physostigmine in laboratory animals and living human beings suffering from neuromuscular diseases, an agent that inhibits cholinesterase [32]. Furthermore, Dotzauer refers to an older study (Jeanselme and Lermoyez, 1885) regarding post-mortem mechanical excitability lasting longer than post-mortem electrical excitability [5]. Cangiano points to an increase of acetylcholine receptors (ACh-receptors) located outside the neuromuscular synaptic membrane that causes hypersensitivity for ACh in denervated skeletal muscles [33]. In a recent study by Yong-il Kim et al., an increased expression of nicotinic Ach-receptors after denervation outside the region of the neuromuscular junction, i.e., neuromuscular end-plate, is suggested as a molecular mechanism leading to hypersensitivity for Ach [34]. Referring to the described mechanisms accountable for hypersensitivity to neurotransmitters, we propose the following two successive pathways for SMR.

In the first hours after death there is enough supply of energy (ATP and glycogen) and intracellular oxygen to meet the demands for depolarisation of the muscle membrane after mechanical stimulation. Because of post-mortem muscle inactivity and the absence of nerve stimuli, extrajunctional Ach-receptors are synthesized. An impact on the muscle leads to the stretching of muscle fibers, which leads to depolarisation of the membrane of the muscle cell at the place of impact due to the release of neurotransmitter ACh. ACh binds to Ach-receptors located at the neuromuscular junction and at extrajunctional locations on the muscle membrane. Because of the absence of physiological nerve stimuli, this depolarisation leads to action potentials remaining local at the place the surface of the muscle was hit. This leads to a local influx of calcium ions from the sarcoplasmic reticulum into the sarcoplasm, initiating a local contraction of the muscle, which can be observed as an idiomuscular pad. After contraction, remaining ATP stimulates the pumping back of calcium ions into the sarcoplasmic reticulum, while other ATP binds to myosin heads, causing crossbridges to detach. The contracted muscle fibers relax. The high level of energy remaining in the muscle cells in the first hours after brain death ensures a relatively fast muscle contraction and relaxation. A contraction of the whole muscle very shortly after death, as described by Zsako and observed by the author once in terms of bending the whole forearm, suggests action potentials that run over the whole surface of the muscle.

In earlier animal studies with steroid-denervated skeletal muscle, it was suggested that inactivation of sodium channels plays a central role in loss of excitability [35,36]. With increasing PMI, oxygen is depleted, and intracellular ATP decreases due to ATP-driven processes such as the pumping of ions through ion channels in membranes. This causes the first pathway to cease because of the failure of sodium channels in the neuromuscular junction. The muscle cells cannot depolarize anymore due to the lack of influx of sodium ions and the associated efflux of potassium ions. Similar effects are associated with hypokalemic periodic paralysis, a rare disease based on genetic disorders of calcium and sodium channels that cause disturbances in the membrane depolarization of muscle cells [37,38]. A post-mortem impact on the muscle leads to pressure on and/or rupture of the membrane of the sarcoplasmic reticulum, which leads to the immediate release of calcium ions into the muscle cell, initiating a muscle contraction (Figure 2).

Lack of ATP leads to function loss of calcium pumps while lowering of the intracellular pH causes membrane dysfunction. This causes calcium to leak from the sarcoplasmic reticulum into the muscle cells and the extracellular matrix. Parts of the sarcomere that contain strong bound crossbridges remain bound because of ATP shortage, while the parts containing weak binding complexes (myosin-ADP-Pi) can bind to actin under the presence of leaked calcium. These weak binding complexes are turned into strong binding complexes (ADP and Pi detach from the myosin head) that cannot detach anymore because of ATP shortage. The contractile properties of muscle decrease because of parts of muscle tissue that cannot contract anymore. However, other parts of muscle tissue might still have the ability to contract because of the presence of sarcomere parts in a different phase of the contraction cycle. Hence, muscle contraction in the case of longer PMI is slower and weaker due to the inability of the muscle membrane to depolarize, ATP depletion, and less formation of crossbridges. As PMI increases, together with a further decrease of ATP, the number of immobile units increases in favor of remaining contractile units, finally causing SMR to disappear.

The course of SMR with increasing PMI results in effects that are similar to exhaustion-related muscle dysfunction, such as during exercise. Skeletal muscle in living organisms shows a decline in performance when used, known as muscle fatigue. Lowering of the intracellular pH due to the accumulation of lactate and hydrogen ions (H+), causing impairment of contractile properties of the muscle fiber, has always been the basic explanation for this phenomenon. Muscle fatigue has been the subject of intensive in vitro studies with skinned muscle fibers derived from laboratory animals, like mice and frogs. Based on these studies, a variety of other mechanisms responsible for muscle fatigue were brought to attention, such as changes in ion concentrations (sodium, potassium, chloride, calcium) that affect generation and conduction of action potentials, ATP depletion causing impairment of ion pumps (sodium, potassium, calcium), accumulation of inorganic phosphate (Pi) which serves as a buffer for calcium ions causing inactivation of the binding sites on actin filaments, failure of calcium release from the sarcoplasmic reticulum and the appearance of oxygen radicals [39,40]. Active oxygen supply to muscle cells ceases when a person dies. Consequently, the oxygen level in muscle cells decreases. This causes a switch from oxidative phosphorylation to anaerobic glycolysis in the still vital muscle cells. Subsequently, energy sources will cease, causing loss of function of ion pumps essential for the excitation-contraction process of muscle cells. Thus, in our opinion, the intracellular changes in muscle cells that are responsible for SMR resemble the intracellular processes responsible for muscle fatigue due to energy exhaustion in the living. The experimental studies on muscle fatigue, which were applied to animal muscle cells [39,40], should be performed on skinned muscle fibers derived from post-mortem human skeletal muscle donated for scientific research in vitro to test the proposed hypothesis of resemblance with muscle fatigue in the living under varying PMI’s. In addition, in combination with the experimental studies on human-skinned skeletal muscle fibers, the course of protein composition in skeletal muscle cells with increasing PMI should be determined [29]. However, it has to be taken into account that there are countries where research with human tissue is not allowed. In accordance with the resemblance of the pig genome with the human genome, the structure of pig skeletal muscle resembles human skeletal muscle. Therefore, skeletal muscle cells derived from pigs would be a justified alternative to perform these investigations under controlled variation of PMI’s [41]. In addition, it will be challenging to perform these experiments with human muscle cells under PMI’s, which are comparable to the daily circumstances in which a forensic physician must operate.

## Figures and Tables

**Figure 1 biomedicines-13-00221-f001:**
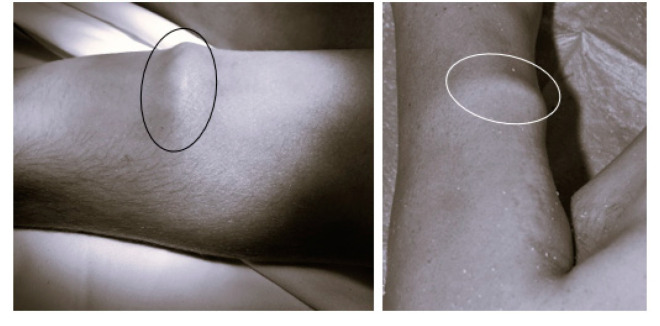
External visible muscular reaction after mechanical stimulation of the musculus biceps brachii (idiomuscular pad).

**Figure 2 biomedicines-13-00221-f002:**
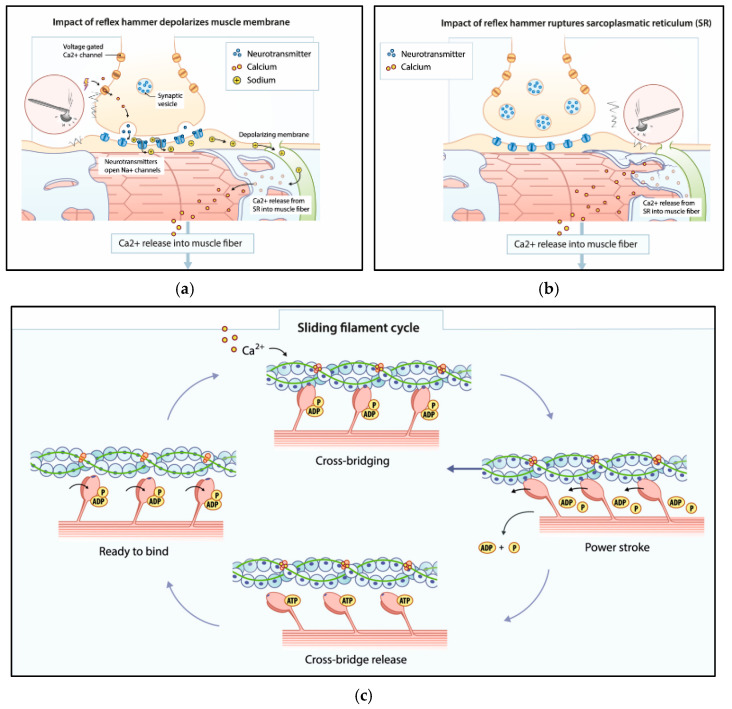
Supravital Muscle Reaction (SMR) after mechanical stimulation. Pathway 1 (**a**): First hours post-mortem; Local depolarization after mechanical stimulation of muscle. Pathway 2 (**b**): Beyond the first hours’ post-mortem; Immediate release of calcium after mechanical stimulation of muscle with the absence of membrane depolarization. (**c**): crossbridge cycle.

## Data Availability

No new data were created or analyzed in this study. Data sharing is not applicable to this article.

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
