# Peer review of "Cell Mechanisms of Post-Mortem Excitability of Skeletal Muscle"

_biomedicines, 2025, doi:10.3390/biomedicines13010221_

Round 1
Reviewer 1 Report (Previous Reviewer 2)
Comments and Suggestions for Authors
Acceptable for publication
Author Response
Reviewer 1 had no suggestions that reguired a response.
Reviewer 2 Report (New Reviewer)
Comments and Suggestions for Authors
The subject of the work is the study of the excitability of post-mortem cellular fibers. The authors have previously published a comprehensive study in which they describe their experience in the practice of mechanical stimulation to induce muscle contractions in deceased subjects in order to establish the time of death. With the present work, they try to hypothesize the cellular mechanisms underlying this phenomenon. The work, although speculative, appears well organized; the hypotheses on the contractile mechanisms after death are plausible and well explained. The literature cited is sufficient considering the small number of works on the subject in question.
Author Response
Reviewer 2 had no suggestions that reguired a response.
Reviewer 3 Report (New Reviewer)
Comments and Suggestions for Authors
Determining the post-mortem interval is one of the most difficult and fascinating challenges in forensic science. In this communication, Stigter and colleagues attempt to shed light on the cellular mechanisms underlying postmortem muscle excitability, a supravital phenomenon whose determination could be integrated for estimating the PMI in daily forensic practice. The topic chosen by the authors is certainly interesting and the paper, although it is only a communication, seems well structured, with a clear division into sections. The presence of figures to illustrate the phenomenon (e.g., Figures 1 and 2) adds visual value and facilitates comprehension. However, I would like to bring to the authors' attention some aspects to be improved, therefore criticisms and suggestions are indicated below.
Although the introduction provides historical context and the phenomenon, it could benefit from a more explicit connection with the proposed hypothesis. For example, it could emphasize how the gap in current knowledge makes research necessary.
Although the theoretical approach is well outlined, an experimental component is missing. It might be useful to include details on how to validate the hypothesis (e.g., experiments with animal muscle fibers or human postmortem tissues, as mentioned).
In addition, the molecular pathway hypothesis is well developed, but empirical data to support it are lacking. For example, to strengthen the "Post-Mortem Muscle Excitability" section, authors could include a reference to this study: doi.org/10.3390/ijms241914627, highlighting how proteomic analysis of postmortem skeletal muscle can provide significant insights into the biochemical changes that influence muscle excitability after death.
Author Response
Comments 1: Although the introduction provides historical context and the phenomenon, it could benefit from a more explicit connection with the proposed hypothesis. For example, it could emphasize how the gap in current knowledge makes research necessary.
Thank you for pointing this out. We have made adjustments in the manuscript, sentence number 52-53 and 56-58:
"Though post-mortem excitability of skeletal muscle is mentioned in textbooks1 and used in daily forensic practice in the Netherlands, knowledge concerning the cell mechanisms of SMR is lacking. Cell mechanisms are the specific intracellular and biochemical processes responsible for post-mortem muscle excitability. The cell mechanisms in muscle cells responsible for muscle contraction in the living is well known and has been subject of extensive in vitro animal studies.10-24 The cell mechanisms in muscle cells responsible for post-mortem muscle contraction are not described in literature. This gap in the current knowledge makes research and a theoretical reflexion on SMR necessary."
Comments 2: Although the theoretical approach is well outlined, an experimental component is missing. It might be useful to include details on how to validate the hypothesis (e.g., experiments with animal muscle fibers or human postmortem tissues, as mentioned).
Thank you for pointing this out. We have made adjustments in the manuscript, sentence number 276-282:
"The experimental studies on muscle fatigue which were applied to animal muscle cells,39,40 should be performed on skinned muscle fibers derived from post-mortem human skeletal muscle donated for scientific research in vitro, to test the proposed hypothesis of resemblance with muscle fatigue in the living under varying PMI’s. In addition, in combination with the experimental studies on human skinned skeletal muscle fibers, the course of protein composition in skeletal muscle cells with increasing PMI should be determined.29 "
Comments 3: In addition, the molecular pathway hypothesis is well developed, but empirical data to support it are lacking. For example, to strengthen the "Post-Mortem Muscle Excitability" section, authors could include a reference to this study: doi.org/10.3390/ijms241914627, highlighting how proteomic analysis of postmortem skeletal muscle can provide significant insights into the biochemical changes that influence muscle excitability after death.
Thank you for pointing this out. We have made adjustments in the manuscript, sentence number 140-146:
- Post-Mortem Muscle Excitability
When brain activity shuts down (clinical death), subsequently nerve stimulation of skeletal muscle stops. However, there is still energy available in muscle cells in terms of glycogen and ATP, while calcium ions are also still present in the muscle cells. Intracellular processes in muscle cells, such as the pumping of ions across membranes through ion channels, do not stop when the brain of the person shuts down at the moment of death. Thus, muscle cells, like other cells in the body, remain vital until exhaustion of energy and post-mortem decrease of the intracellular pH due to anaerobic glycolysis and build-up of carbon dioxide (right shift in the acidic base equilibrium) leads to death of the muscle cell. This period of temporary survival of muscle cells after brain death is fundamental for the post-mortem excitability of skeletal muscle.
After clinical death, as long ATP is available, calcium is still actively pumped back into the sarcoplasmic reticulum. The intracellular calcium concentration decreases, which leads to deactivation of active sites on the actine filaments. Myosin heads detach from the actin filament while other myosin heads cannot bind to actin because of the presence of ATP on the myosin head. Other crossbridge formations are weak bounded crossbridges (ATP hydrolysed in ADP and Pi) that detach because of shortage of calcium ions, where there are also strong bounded crossbridges, remaining in a transiently rigor state (myosin heads without ADP and Pi) despite the presence of a lower concentration of calcium ions.
Post-mortem mechanical stimulation of skeletal muscle can result in a visible and manually detectable contraction and relaxation. As described by Dotzauer et al. and observed in daily practice, dependent on the PMI, the strength, magnitude and velocity of post-mortem contraction and relaxation of skeletal muscle, observed after mechanical stimulation, can be divided into three levels of degree. In the first 1.5 to 2.5 hours after death a contraction of the whole muscle can be observed which corresponds with the muscle contraction observed by Zsako in the 1910s.3 This is followed by a strong and reversible idiomuscular pad (4 – 5 hours post-mortem; Figure 1). At the third level only a weak idiomuscular pad can be observed which may persist for a long period.5,25,26 Furthermore, with longer PMI autolysis based on tissue decay by enzymatic degradation processes, undermines the ability of muscle to contract due to loss of functional tissue.27, 28 Finally, muscle contraction can no longer be observed after mechanical stimulation.
An interesting and promising field of scientific development is the area of forensic proteomics, in which proteomic analysis of post-mortem skeletal muscle can provide insights into the biochemical changes that influence muscle excitability after death. Marrone et al. described nine proteins showing a decreasing or increasing trend of concentration in the muscle cell with increasing PMI. Among the proteins decreasing with longer PMI is the so-called Muscle-restricted coiled-coil protein (MURC), which is present in the sarcomere of skeletal muscle. MURC is involved in myogenesis of skeletal muscle cells.29
This manuscript is a resubmission of an earlier submission. The following is a list of the peer review reports and author responses from that submission.
Round 1
Reviewer 1 Report
Comments and Suggestions for Authors
The article reports a novel interpretation of muscle contraction post mortem. The topic is interesting and appropriate for the journal, but I feel not so in the current form unfortunately.
Major concerns:
For me a communication article would normally address an exciting new development as part of a larger study - I don't see any evidence of that here unless I have missed something. A hypothesis is presented with no supporting evidence in the form of an interpretation of pre-existing interpretations.
There is much literature on animal models of post mortem muscle contraction that is not mentioned here, why is the unreferenced image of a human arm so pertinent?
Minor concerns:
The authors would have to explain to me what 'Celmechanisms' actually means as I'm afraid I am not familiar with the term and it is not explained in the text.
Figures 2 and 3 are far too similar to the point of being repetitive for no added benefit.
Author Response
I. Comments 1: "For me a communication article would normally address an exciting new development as part of a larger study - I don't see any evidence of that here unless I have missed something. A hypothesis is presented with no supporting evidence in the form of an interpretation of pre-existing interpretations."
Reply: This paper is part of a larger study concerning the applicability of post-mortem excitability of skeletal muscle for estimating the post-mortem interval (PMI) in daily forensic practice in the Netherlands. We have adjusted this in the revised manuscript by mentioning the larger study with the enclosed references in the section "Introduction", page 1 and 2; sentence number 42 to 49.
We performed a thorough literature search about this phenomenon. Besides older studies, that mainly describe the phenomenon in general and contain only brief remarks about the possible underlying mechanims. We did not find older and recent studies about this topic focussed on the cellular processes that could be responsible for this post-mortem muscle excitability. Thus, although the instrument of post-mortem excitability of skeletal muscle is mentioned in textbooks (for instance Madea et al., 2016), the specific cellular processes acounting for muscle excitability post-mortem are not well defined.
II. Comments 2: "There is much literature on animal models of post mortem muscle contraction that is not mentioned here, why is the unreferenced image of a human arm so pertinent?"
Reply:
-We have included more literature about animal studies concerning the cellular mechanisms that are being held accountable for muscle contraction (reference numbers 10 to 24 in teh revised manuscript). No literature has been found that deals specifically with post-mortem muscle excitability
-Image human arms: The photographs were produced during investigation in the own forensic practice in collaboration with forensic investigators of law inforcement. The figure in the manuscript has been adjusted to a smaller size.
III. Comments 3: "The authors would have to explain to me what 'Celmechanisms' actually means as I'm afraid I am not familiar with the term and it is not explained in the text."
Reply: With the term 'celmechanisms' we refer to the specific intracellular and biochemical processes responsible for post-mortem muscle excitability. We adjusted this in the manuscript in the section Abstract, page 1; sentence number 19-21 and the section "Introduction", page 2; sentence number 51-54.
IV. Comments 4: "Figures 2 and 3 are far too similar to the point of being repetitive for no added benefit."
Reply: we have combined the two figures to one figure, with a more clearer explanation of the chronology of both pathways (page 5 of the revised manuscript).
Reviewer 2 Report
Comments and Suggestions for Authors
The authors' approach is acceptable. Their hypothesis is based on previous knowledge, written in clear terms and refers to concrete and verifiable facts.
The authors mentioned in the Abstract a field study in the Netherlands to investigate the applicability of muscle excitability by mechanical stimulation for estimating the post-mortem interval in daily forensic practice. And it is a very relevant justification for the present literature review. However, it is striking that the study is not mentioned in the Introduction section.
In addition, It is striking that the term ‘celmechanisms’ does not appear throughout the text of the article. Though it is understood, the authors should mention it in the article text and briefly explain its meaning.
The authors claim (at the bottom of page 6) that experimental studies applied on animal muscle cells should be applied on human skeletal muscle cells in vitro to test their hypothesis. Thus, authors are encouraged to include a very brief proposal of what might be accomplished in the future (based on 24 & 25 references, for instance).
Author Response
I. Comments 1: "The authors mentioned in the Abstract a field study in the Netherlands to investigate the applicability of muscle excitability by mechanical stimulation for estimating the post-mortem interval in daily forensic practice. And it is a very relevant justification for the present literature review. However, it is striking that the study is not mentioned in the Introduction section."
Reply: This paper is part of a larger study concerning the applicability of post-mortem excitability of skeletal muscle for estimating the post-mortem interval (PMI) in daily forensic practice in the Netherlands. We have adjusted this by mentioning the larger study with the enclosed references in the section "Introduction", page 1 and 2; sentence number 42-49.
II. Comments 2: "In addition, It is striking that the term ‘celmechanisms’ does not appear throughout the text of the article. Though it is understood, the authors should mention it in the article text and briefly explain its meaning. "
Reply: Cell mechanisms are the specific intracellular and biochemical processes responsible for post-mortem muscle excitability. We adjusted this in the manuscript in the section "Abstract", page 1; sentence number 19-21 and the section "Introduction", page 2; sentence number 51-54.
III. Comments 3: "The authors claim (at the bottom of page 6) that experimental studies applied on animal muscle cells should be applied on human skeletal muscle cells in vitro to test their hypothesis. Thus, authors are encouraged to include a very brief proposal of what might be accomplished in the future (based on 24 & 25 references, for instance)."
Reply: We have described our general thoughts about these possible future studies and also the problems that could be faced using human muscle tissue. We have adjusted this in the revised manuscript in the section 'Hypothesis", page 6: sentence number 269-278.
Reviewer 3 Report
Comments and Suggestions for Authors
In this paper, the authors explore possible mechanisms to explain post-mortem muscle motion in response to mechanical stimulation.
The concept behind the paper is quite interesting and original, but the hypotheses posited in the manuscript are somewhat naïve and the physiological evidence on which the assay (because that’s what it is) is somewhat weak. While I understand that this is an underexplored field, the references included in the study are very outdated for the most part.
When exploring the action of myosin II in live muscle, the authors have elected to cite generic reviews, instead of quoting recent reviews from significant researchers in the field, including Anne Houdusse, Michael Geeves and others. This needs to be addressed. Since this reviewer is not any of these researchers, I don’t feel guilty about asking the authors to bring up better and more significant articles and reviews.
Some concepts are mentioned but poorly explained. For example, the authors base most of their argumentation on Cannon and Rosenblueth’s law, but the science underlying this, or whether this is still a majority viewpoint in the field, are not really discussed. Another issue is the formation of the idiomuscular pad, which is mentioned and shown, but the mechanisms underlying its formation are hardly mentioned.
In general, the authors need to extend their arguments, to make their paper a more compelling read.
Conversely, the figures are excellent.
Author Response
I. Comments 1: "The concept behind the paper is quite interesting and original, but the hypotheses posited in the manuscript are somewhat naïve and the physiological evidence on which the assay (because that’s what it is) is somewhat weak. While I understand that this is an underexplored field, the references included in the study are very outdated for the most part."
Reply: After a thorough literature research, we have found many animal studies concerning cellular mechanisms/processes that are held accountable for muscle contraction. These studies do not deal with the post-mortem situation. Indeed, several studies dealing with the post-mortem excitability of skeletal muscle were performed a long time ago, in which the possible intracellular processes that could be responsible for this phenomenon are only briefly described. Thus, though the phenomenon is described in textbooks (for instance Madea et al., 2016) and is used in practice as an additional tool for estimating PMI, a clear explanation for this phenomenom from a intracellular, biomechanical point of view is lacking. So, in order to pose a hypothesis about the cellular mechanisms that could play a role in post-mortem excitability of skeletal muscle, we think it is justified to use the situation in case of living organisms as a starting point.
II. Comments 2: "When exploring the action of myosin II in live muscle, the authors have elected to cite generic reviews, instead of quoting recent reviews from significant researchers in the field, including Anne Houdusse, Michael Geeves and others. This needs to be addressed. Since this reviewer is not any of these researchers, I don’t feel guilty about asking the authors to bring up better and more significant articles and reviews."
Reply: a valuable comment of the reviewer; We have included (after studying these papers) the studies performed by the mentioned investigators. i.e. reference numbers in the revised manuscript: 15-24. Moreover, we performed an additional research of the literature using the keywords mentioned in these studies. We found no additional studies dealing with post-mortem excitability of skeletal muscle.
III. Comments 3: "Some concepts are mentioned but poorly explained. For example, the authors base most of their argumentation on Cannon and Rosenblueth’s law, but the science underlying this, or whether this is still a majority viewpoint in the field, are not really discussed. Another issue is the formation of the idiomuscular pad, which is mentioned and shown, but the mechanisms underlying its formation are hardly mentioned. "
Reply: Cannon and Rosenblueth; Although in a textbook the term 'Cannon and Rosenblueth's law' is used, we have decided to delete the term 'law', which in our opinion is more appropiate with the outcome in later studies; We have included more recent studies that deal with the possible mechanisms responsible for the changes at the membrane of muscle cell, when nerve stimulation stops (denervation). In those recenter studies more specific explanations are mentioned. Concerning this comment we have made adjustments in the revised manuscript in the section 'Hypothesis' page 4; sentence number 160-172 and 193-194.
Idiomuscular pad: As mentioned in the reply on Comment I, a clear explanation for this phenomenom from a intracellular, biomechanical point of view is lacking, although the phenomenon is described in textbooks and is being used in practice as a additional tool for estimating PMI. The mechanisms that in our opinion could be responsible for this muscle reaction is described in the section concerning the '3. Post-mortem Muscle excitability' and '4. Hypothesis', in combination with the included figures. Until now, it is an unexplored phenomenon from a cellular point of view, besides that it can be observed after hitting a muscle with a reflex hammer (mechanical stimulation).